# Effects of Hyperbaric Oxygen Therapy in Children with Severe Atopic Dermatitis

**DOI:** 10.3390/jcm10061157

**Published:** 2021-03-10

**Authors:** Judyta Mews, Agata Tomaszewska, Jacek Siewiera, Sławomir Lewicki, Karolina Kuczborska, Agnieszka Lipińska-Opałka, Bolesław Kalicki

**Affiliations:** 1Department of Pediatrics, Nephrology and Allergology, Military Institute of Medicine, Szaserów 128 Str., 04-141 Warsaw, Poland; awawrzyniak@wim.mil.pl (A.T.); kmorska@wp.pl (K.K.); alipinska@wim.mil.pl (A.L.-O.); kalicki@wim.mil.pl (B.K.); 2Clinical Department of Hyperbaric Medicine, Military Institute of Medicine, 04-141 Warsaw, Poland; jsiewiera@wim.mil.pl; 3Department of Regenerative Medicine and Cell Biology, Military Institute of Hygiene and Epidemiology, 04-141 Warsaw, Poland; slawomir.lewicki@wihe.pl; 4Faculty of Medical Sciences, Kazimierz Pulaski University of Technology and Humanities, 26-610 Radom, Poland

**Keywords:** atopic dermatitis, hyperbaric chamber, immune system, SCORAD, children

## Abstract

In the course of atopic dermatitis (AD), the overactivity of the immune system, associated with predominant Th2 lymphocyte responses, is observed, which leads to an increased inflammatory reaction. Cases of a severe course of atopic dermatitis lead to the search for new therapeutic options. The aim of this study was to assess the effects of hyperbaric oxygen therapy (HBOT) treatment for severe cases of AD in children. A total of 15 children with severe AD underwent therapy. The influence of HBOT on the clinical course of AD and immunomodulatory effect of the therapy was analyzed by the SCORAD and objective SCORAD (oSCORAD) scales and by determining the serum concentration of immunological parameters (blood: nTreg lymphocytes, CD4+CD25highCD127-FOXP3+, NKT lymphocytes CD3+, CD16/56+, and serum: total IgE, cytokines IL-4, IL-6, and IL-10, before and after the 30-day treatment cycle). The study showed a significant effect of the therapy on the improvement of the skin condition. In all children, a reduction in the extent and intensity of skin lesions, reduction of redness, swelling, oozing/crusting, scratch marks and skin lichenification after HBOT was observed. Patients also reported a reduction in the intensity of pruritus and an improvement in sleep quality after therapy. In all children, a statistically significant decrease in the serum level of IgE was observed. However, no statistically significant changes in the blood levels of IL-4, IL-6 and IL-10, as well as the percentage of CD4^+^CD25^high^CD127^−^FOXP3^+^ Treg and NKT lymphocytes, were found. In conclusion, the use of hyperbaric therapy has a positive impact on treatment results in children with a severe course of atopic dermatitis.

## 1. Introduction

Atopic dermatitis (AD) is a chronic inflammatory skin disease with periods of exacerbation and remission associated with skin barrier dysfunction. A typical feature of this chronic dermatosis is the characteristic morphology and localization of skin lesions, persistent and recurrent pruritus, as well as skin lichenification [1,2,3]. In the course of the disease, quality of life is significantly reduced. Atopic dermatitis generally develops in early childhood, before the age of one. In 45% of AD cases, the first symptoms occur before six months of age [3,4,5]. The pathogenesis of the disease is complex and not fully understood.

The incidence of AD is caused by genetic factors (related, among others, to filaggrin mutations), environmental factors (lifestyle, diet) and immune dysregulation [5,6,7]. One of the characteristic phenomena of AD is dysregulation of the Th1/Th2 response, in which CD4 T cells differentiated into the Th2 lineage are facilitated and proliferation of the Th1 lineage is impaired. As a result of the Th1/Th2 balance shift, the profile of produced cytokines in the blood changes (increased production of IL-4, IL-5 and IL-13). An increased concentration of Th2 cytokines causes enhanced immunoglobulin E production. IgE-mediated antigen presentation plays an important role in the pathogenesis of atopy; however, the serum IgE levels do not always correlate with the severity of symptoms [7].

Recent studies in humans and animal models demonstrate that the keratinocytes of atopic skin lesions synthesize and release thymic stromal lymphopoietin (TSLP), the expression of which is strongly correlated with the severity of disease activity. TSLP acts as a potent stimulator of the cytokines of the Th2 profile and induces the proliferation, differentiation and activation of mast cells. TSLP may work as a major mediator to initiate and maintain an inflammatory response in the skin of patients with AD. Studies also indicate the role of TSLP in the pathogenesis of perceived pruritus in atopic dermatitis [8,9].

The treatment of atopic dermatitis has two main aspects: relieving symptoms and eliminating the identified specific allergens from the patient’s environment, which cause exacerbations of skin lesions. The basic element of pharmacotherapy in AD is proper skin care with emollients. Symptomatic treatment includes: glucocorticosteroids (used both topically and generally, especially in exacerbations), antihistamines, calcineurin inhibitors (topical pimecrolimus and tacrolimus), phototherapy and photochemotherapy. In severe or refractory cases, immunosuppressive drugs are used (cyclosporine and methotrexate). There are also attempts to use biological treatment with monoclonal antibodies (omalizumab, dupilumab) [10,11,12]. Cases of a severe course of atopic dermatitis that do not respond to available treatment methods lead to the search for new, more effective therapeutic options.

A new method of treatment of atopic dermatitis is hyperbaric oxygen therapy (HBOT). Although the results of the numerous studies enable its wide use in medicine, there are few reports on its use in children with AD [10,11,12]. Thus, we sought to assess the efficacy of hyperbaric oxygen therapy (HBOT) in the treatment of severe AD in children. The available studies concern the use of HBOT in the treatment of, among others, carbon monoxide poisoning, chronic wounds and decompression sickness. Hyperbaric oxygen therapy is a therapeutic method involving the use of 100% oxygen under conditions of elevated pressure, i.e., at a pressure of at least 2 ATA (absolute atmospheres) for a period of at least 60 min. The therapy is conducted in specially constructed rooms called hyperbaric chambers [10,11,12].

The aim of this study was to evaluate the treatment effectiveness of severe cases of AD in children with the method of hyperbaric oxygen therapy (HBOT).

## 2. Materials and Methods

### 2.1. Patients

The study was designed as a prospective study. It was approved by the Bioethics Committee at the Military Medical Chamber No. 159/18.

A total of 15 children with severe atopic dermatitis (8 boys and 7 girls, aged 3–16) were enrolled in the study. AD was confirmed by the positive results of a skin prick test in all participants.

Each time, before child qualification for treatment in a hyperbaric chamber, thorough medical history and physical examinations were performed. The inclusion and exclusion criteria for the study are presented in Table 1.

Before the oxygen hyperbaric therapy initiation, the patients, apart from complex topical therapy, were periodically administered orally with prednisone (3 patients) and cyclosporine (2 patients). However, during HBO treatment, patients used only topical treatment (skin care with emollients). Possible systemic glucocorticotherapy and/or cyclosporine treatment was discontinued at least 6 months before HBOT initiation. These assumptions eliminated the effect of the pharmacological suppression of the immune system. Ad hoc antihistamine intake (1st and 2nd generation) was also allowed in order to reduce the itching sensation.

All patients completed a 30-day treatment cycle in a hyperbaric chamber.

### 2.2. Severity of Skin Lesions

The clinical disease activity was analyzed by the SCORAD questionnaire and its modification, objective SCORAD (oSCORAD).

The SCORAD method was used to assess objective symptoms (the extent and severity of skin lesions) and subjective symptoms (severity of pruritus and sleep disorders). The extent of the affected area was estimated using the rule of nines. Six symptoms were assessed to determine the severity of skin lesions—skin dryness, redness, swelling, oozing/crusting, scratch marks and lichenification—on a 4-point scale, from 0 to 3. Each grade was assigned an appropriate number of points: 0 points for no lesions and 3 points for the most severe cases, respectively. Skin dryness was assessed in the area not affected by the disease, and the remaining symptoms were assessed in the most representative areas. The severity of pruritus and sleep disorders was each scored by the patient on a visual analog scale (ranging from 0 to 10) as the mean value of the previous 3 days and 3 nights. In the case of children under the age of 7, this was performed by the child’s guardian. On the basis of the obtained results, AD was classified into mild (<25 points), moderate (25–50 points) or severe (>50 points). The maximum score on the SCORAD scale is 103 points.

In order to eliminate subjective symptoms, i.e., pruritus and sleep disorders, clinical AD activity was assessed simultaneously with the objective SCOARD (oSCORAD) questionnaire. The severity of the disease on the oSCORAD scale was rated as mild (<15 points), moderate (15-40 points) or severe (>40 points). The maximum score on the oSCORAD scale is 83 points.

SCORAD and oSCORAD (Assessment of the Severity/Activity of the Disease) Were Assessed up to 24 h before Blood Collection for Laboratory Tests, Always up to 24 h before the Initiation of HBO Therapy and no Later than 48 h after the End of a 30-day Cycle of Hyperbaric Oxygen Therapy. Blood Collection for Laboratory Tests Was Performed up to 24 h before the Initiation of HBO Therapy and up to 48 h after the End of a 30-Day Cycle of the Therapy. 

### 2.3. Immunological Parameters

In all treated patients, the serum concentration of total IgE antibodies and cytokines IL-4, IL-6 and IL-10, as well as the percentage of CD4^+^CD25^high^CD127^−^FOXP3+ Treg and NKT peripheral blood lymphocytes, was determined. The analyses were performed before and 30 days after hyperbaric oxygen treatment

For serum analysis, the collected blood samples were allowed to clot. After 20 min (room temperature), the blood was centrifuged (2000× *g*, 20 min), and the serum was collected. Immediately after the collection, total IgE concentration was determined. For cytokine examination, the serum was portioned and stored at −80 °C until the analysis. For immune cell analysis, vein blood was collected on EDTA. Immediately after isolation, nTREg and NKT lymphocyte analysis was performed.

Total IgE concentration was determined using the fluoroimmunoenzymatic method with the Unicap100 or Phadia100 fluorimeter manufactured by HVD.

Cytokine concentrations were determined by a cytometric bead array (CBA, Human, Th1/Th2/Th17, BD Bioscience, Warsaw, Poland) on the FACSCalibur flow cytometer, according to the manufacturer’s procedure. In brief, the following were added to the cytometric tubes: 50 μL of capture beads, 50 μL of the detection reagent, 50 μL of the test serum (test sample), 50 μL of the known concentration of cytokines (standard curve: 20–5000 pg/mL) or 50 μL of blank (blind sample). The samples were then incubated for 3 h, washed and centrifuged (200× *g*, 5 min). Finally, the supernatant was collected from each sample, and 300 μL of washing buffer was added.

Cytokine concentrations were determined on the FACSCalibur flow cytometer, and the results were analyzed with the use of FCAP Array™ Software (BD Bioscience, San Jose, CA, USA). The results were converted into the value obtained from the standard curve and presented as the mean concentration of the tested cytokine (pg/mL) ± SD.

The percentage of nTreg (antibody against CD4, CD25, CD127 and FOXP3) and NKT lymphocytes in the blood (IMK assay) from BD Bioscience (Warsaw Poland) were analyzed as previously described [13]. Analyses were performed on the FACSCalibur flow cytometer (USA). The results were presented as the mean percentage ± SD.

### 2.4. Hyperbaric Oxygen Therapy

The course of the HBOT included three 20 min cycles, during which oxygen was administered under hyperbaric conditions, and 5 min air breaks between the cycles. The aim of the breaks was to increase the safety of the procedure and to reduce the possible side effects related to oxygen therapy. The total time of breathing with hyperbaric oxygen was 60 min for each session. During the session, the patients underwent two 10 min periods of compression to the desired ATA pressure in the patient’s environment and decompression, during which the participants were breathing air. These periods were performed at the beginning and at the end of the procedure, respectively (Figure 1). The procedure was performed once a day for 30 days, considering the individual tolerance of the procedure by each patient.

The treatment protocol performed in a hyperbaric chamber included compressing the patient in the air atmosphere to the maximum operating depth (MOD) value of 15 m H_2_O, which corresponds to the value of 2.5 ATA (~250 kPa), with an individually tolerated velocity no faster than 1.5 m H_2_O/min. This protocol provides for a safety stop during decompression.

Depending on the age and the cooperation degree with the staff, the patients were subjected to the therapy alone or with the assistance of a legal guardian as a medical attendant. The attendant did not breathe the hyperbaric oxygen for therapeutic purposes but performed oxygen denitrogenation with the use of a mask-type respiratory system 10 min before the start of decompression (together with the chamber medical staff). The aim of pre-decompression denitrogenation was to minimize the risk of decompression sickness. Children subjected to HBOT procedures were treated mainly with the use of “S”-size “mask” respiratory systems. In the case of patients under 7 years of age, or those who did not tolerate the mask, a hood-type oxygen helmet was used with a continuous flow of oxygen of 15 L/min.

### 2.5. Statistical Analysis

Before the analysis, the data were initially verified with the use of a normal probability plot and finally with the Kolmogorov–Smirnov and Lilliefors normality test. The Student’s *t*-test was used to compare normally distributed variables and the Mann–Whitney U test to compare variables of which the distribution was significantly different from the Gaussian distribution. Every time, the probability value of *p* < 0.05 was considered statistically significant. StatSoft, Inc. (2014, Cracow, Poland) and Statistica 12 were used for analysis.

## 3. Results

### 3.1. Characteristics of the Study Group

Fifteen patients aged 3–16 years (7 girls and 8 boys) qualified for HBOT. The characteristics of the study group are presented in Table 2 (Table 2).

The following coexisting atopic diseases were observed in children undergoing HBOT: allergic rhinitis (3 patients), inhalation and/or food allergy (12 patients). Two out of fifteen patients had a positive family history of allergy. Due to severe itching, 11 children used antihistamines during hyperbaric sessions. All children completed a 30-day HBOT course.

### 3.2. Skin Condition

A statistically significant clinical improvement in the skin condition was observed in all children subjected to the therapy, which was assessed using SCORAD and oSCORAD questionnaires (Figure 2 and Figure 3). Clinical assessment of the AD activity was assessed before the initiation of HBO therapy and after the end of the 30-day cycle of the therapy.

The beneficial local effect of HBOT was observed in reducing the extent of skin lesions (Figure 4).

There was also a reduction in skin dryness, erythema severity, the presence of edema, exudates, scratch marks and skin lichenification. Patients also reported a reduction in the intensity of pruritus and an improvement in sleep quality after HBOT (Table 3).

In conclusion, after the 30-day cycle of HBO treatment, a beneficial clinical improvement in the skin condition was observed. There was also a reduction in the intensity of pruritus and an improvement in sleep quality in all patients after HBOT.

### 3.3. Immunological Parameters

In all children, in terms of the assessed immunological parameters, a statistically significant decrease in the serum total IgE concentration was observed after the cycle of hyperbaric exposure was completed. However, there was no effect of HBOT on the serum concentration of the cytokines (IL-4, IL-6 and IL-10). No statistical significance was proved in the percentage changes of the Treg CD4^+^CD25^high^CD127^−^FOXP3+ and NKT peripheral blood lymphocytes before and after the 30-day cycle of HBOT (Table 4).

In conclusion, there was no significant influence of HBOT on the concentrations of IL-4, IL-6 and IL-10 cytokines in the blood serum in the examined children before and after the therapy. There was no evidence of a change in the profile of cytokine production after the treatment in a hyperbaric chamber. There were no changes in the percentages of the NKT and TregCD4^+^CD25^high^CD127^−^FOXP3^+^ lymphocytes in the peripheral blood of patients with severe AD before and after the 30-day cycle of HBO therapy.

## 4. Discussion

Atopic dermatitis is one of the most common inflammatory skin diseases in children. The course of AD is chronic and recurrent. The predominant symptom of the disease is intense pruritus, which occurs not only during the day but also at night [14]. The consequences of such a course of the disease are insomnia, irritability, stress and social functioning disorders, both of a sick child and their parents. The more severe the course of AD, the greater its impact on the deterioration of the quality of life of the whole family [15]. The clinical problem that atopic dermatitis poses for modern medicine prompts us to search for new methods of treatment, which would alleviate the course of the disease or avoid its exacerbations. These needs are met, e.g., by hyperbaric therapy [16,17,18,19,20,21,22,23,24].

In our study, it was shown that during hyperbaric oxygen therapy, all children with severe atopic dermatitis experienced a significant clinical improvement in their skin condition. After the 30-day treatment cycle in a hyperbaric chamber, there was a reduction in the extent and intensity of skin lesions, as well as a reduction of redness, swelling, oozing/crusting, scratch marks and skin lichenification. Moreover, patients reported a noticeable reduction in the intensity of pruritus and an improvement in sleep quality. The beneficial clinical effect of HBOT was achieved only with the use of systematic topical treatment emollients. In the course of hyperbaric therapy, no topical corticosteroids or other standard treatment methods were used. Similar results were obtained by Olszański et al. in their study concerning the treatment of AD in adults. All 10 patients after treatment with hyperbaric oxygenation experienced a significant local improvement in the field of skin lesions (in the opinion of a dermatologist who previously referred these patients to participate in hyperbaric exposures), while the patients themselves declared a noticeable reduction in perceived itching [25].

In our study, we tried to answer the question regarding the impact of HBOT on the immune system of children suffering from severe atopic dermatitis. The immunomodulatory effect of HBOT is nonselective and affects the function of both T and B lymphocytes. In the experiments performed on animals, it was proved that during exposure to hyperbaric oxygenation (HBO), various types of immune responses are inhibited [26,27]. The molecular mechanisms underlying the immunomodulatory effects of hyperbaric oxygen therapy on the organism are known only partially. Based on the available literature, it can be concluded that the fundamental process leading to AD is an imbalance between Th1 and Th2 lymphocytes, as well as the reduced number and/or impaired function of regulatory T cells. Many factors influence the differentiation of lymphocytes in the Th1 and Th2 lines. Available studies show that natural killer T cells (NKT) may also be initiators of an allergen-specific immune response similar to dendritic cells (DCs). Under physiological conditions, NKT cells are capable of producing INF-γ in response to stimulation by dendritic cells. It has been observed that in people with allergies, the number of circulating NKT in the blood serum is reduced. This leads to a decrease in the production of INF-γ, which is a natural inhibitor of the Th2-dependent response [28,29].

The results of studies conducted around the world also show a significantly lower percentage of natural regulatory T cells (nTreg), defined as CD4+/CD25high/CD127low/FoxP3+, in the blood serum of children with AD than in the healthy population [30,31,32,33,34,35,36,37].

Many publications emphasize the anti-inflammatory effect of HBOT in the treatment of diseases with an intense inflammatory reaction, e.g., psoriasis, difficult-to-heal wounds (such as a diabetic foot), ulcerations or skin burns [38,39,40]. These studies confirm that by increasing the oxygen availability at the cellular level, HBOT elevates the production of reactive oxygen species (ROS) in the body’s tissues [40,41,42,43]. With the participation of immunomodulatory molecules indoleamino-2,3-dioxygenase (IDO) and hypoxia-induced factor 1α (HIF-1α), HBOT exerts a key influence on the function and differentiation of regulatory T lymphocytes (Treg) [44,45,46,47]. Depending on the tissue concentration of ROS and the expression of selected regulatory molecules, Treg lymphocytes differentiate into Treg CD4^+^CD25^high^CD127^−^FOXP3^+^ (in the case of an increased concentration of IDO) or Th17 (in the case of the predominance of HIF-1α expression) [48,49]. The differentiation in one of the above-mentioned directions causes the simultaneous inhibition of the differentiation in the opposite direction. Tissue hyperoxygenation through the increased skin concentration of ROS alleviates the course of AD in experimental animals [50]. It was proved that an increased IDO expression and a decreased HIF-1α level in skin lesions in mice treated with HBOT influence the differentiation of Treg cells, with a significant predominance of CD4^+^CD25^high^CD127^−^FOXP3^+^Treg in diseased tissues. In biopsy samples of skin lesions collected from areas with high disease intensity, the number of these cells was significantly higher after HBOT treatment [50].

Taking into account the limitation associated with obtaining skin biopsies in children with AD, in our study, in addition to the percentage of CD4^+^CD25^high^CD127^−^FOXP3^+^ Treg and NKT cells in the peripheral blood, the concentration of IL-10 in blood serum was also assessed (cytokines secreted by induced Treg cells). Moreover, in the study, we investigated the changes in the Th2 cytokine profile–IL-4 (increased production in allergic reactions) and IL-6 (increased production in inflammatory reactions) in order to assess the influence of HBOT on the severity of inflammation in patients with AD.

In the study, however, it was not possible to confirm the above-mentioned hypothesis about the beneficial effect of HBOT on the Th2 cells in relation to the Th1 cells and the cytokines produced by them. In children undergoing the therapy, mean concentrations of Th2 cytokines (IL-4 and IL-6) before and after the 30-day cycle of HBOT were similar. A relatively constant percentage of CD4^+^CD25^high^CD127^−^FOXP3^+^ Treg lymphocytes and NKT cells, as well as relatively constant concentrations of IL-10 produced by induced Treg before and after HBOT, was demonstrated.

The lack of statistically significant differences in the concentration of cytokines before and after HBOT may result from the too-short duration of the therapy for changes in the serum concentrations of the analyzed immunological parameters. Perhaps the immune profile should be reassessed a few months after the end of the therapy or the number of HBOT sessions should be increased (>30). Greater differences could then be observed.

B cells, as antigen-presenting cells and T-cell activators in allergic inflammation, play a major role in the secretion of specific antibodies. B cells are a source of IgE antibodies that initiate an allergic reaction by the attachment of an allergen, which then leads to the degranulation of mast cells and the release of inflammatory mediators [51,52,53].

In our study, a statistically significant decrease in the serum concentration of total IgE after the 30-day treatment cycle was shown, which proves the influence of HBOT on the synthesis of these specific immune proteins (IgE) by B cells. This confirms the conclusions resulting from the previously mentioned studies on the influence of HBOT on the reduction of the inflammatory reaction in the course of AD. In addition, the reduction in total IgE levels may be related to the reduction in pruritus observed in patients after HBOT therapy. For many years, the source of itching in AD has been seen in the mast cells-histamine axis, but recent studies show that IgE-R + basophil–neuronal interactions play a key role here [8,9].

The study showed a clear beneficial local effect of HBOT on the clinical skin condition and the alleviation of the course of AD. Therefore, we confirmed the conclusions drawn from the research studies cited in the article regarding the influence of tissue hyperoxygenation on the functioning of the immune system cells directly in the diseased skin. In the study, we did not assess immune cell activity and cytokine concentrations (IL-4, IL-6 and IL-10) directly in the skin biopsies. Therefore, due to the different methodology in the experiments mentioned in this article, it is impossible to unequivocally compare the results obtained by other researchers with our results.

It is worth mentioning that HBOT is a highly safe method. Researches showed that the risk of symptoms of oxygen toxicity in the treated patients is low [54,55].

## 5. Conclusions

The use of hyperbaric therapy in children with a severe course of atopic dermatitis has a positive impact on the treatment results of this disease. It reduces the severity of skin lesions, as well as its dryness, the presence of redness, swelling, oozing/crusting, scratch marks and lichenification. HBOT has a positive effect on the reduction of pruritus intensity and improves sleep quality in patients with severe AD. This has a significant impact on improving the quality of life of these patients.Hyperbaric oxygen therapy may be a therapeutic option for some patients with atopic dermatitis, especially in severe cases that are resistant to the standard methods of treatment.The study did not show a clear effect of hyperbaric oxygen therapy on the functioning of the immune system. Immunological parameters presented in this study are not sufficient to discuss all of the molecular mechanisms underlying the immunomodulatory effects of HBOT.

However, due to the promising results of the local effects of the therapy in the skin condition of patients with AD, it is necessary to conduct further studies in this area on a larger group of patients and randomized patients.

The study was approved by the Bioethics Committee at the Military Medical Chamber No. 159/18.

## Figures and Tables

**Figure 1 jcm-10-01157-f001:**
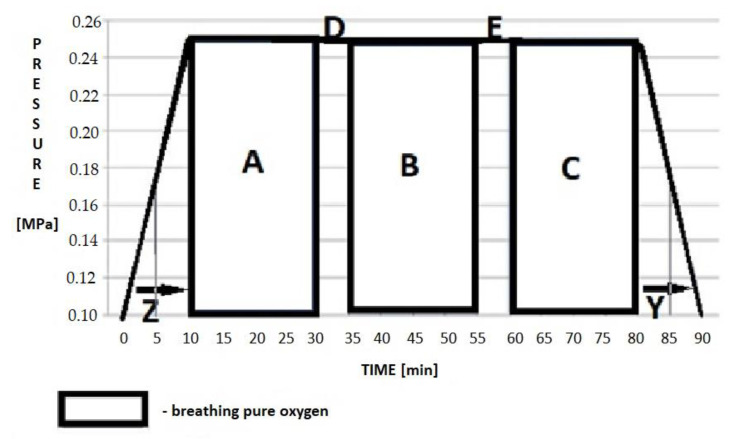
Treatment profile in a hyperbaric chamber: (A–C) time during which the patient was breathing pure oxygen under hyperbaric conditions with the use of a mask or helmet; (D,E) air brakes aimed at breathing air contained in a hyperbaric chamber without the use of individual oxygen devices. Z—time to full compression. Y—decompression time.

**Figure 2 jcm-10-01157-f002:**
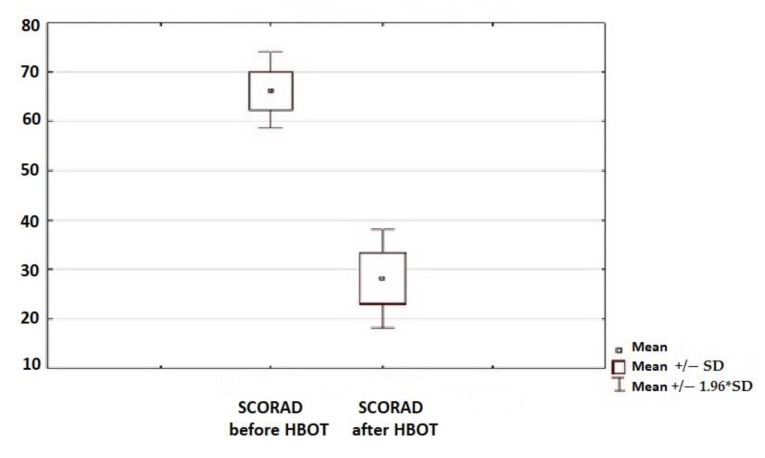
Assessment of the disease activity before and after hyperbaric oxygen therapy (HBOT) with the use of the SCORAD method (*p* < 0.05).

**Figure 3 jcm-10-01157-f003:**
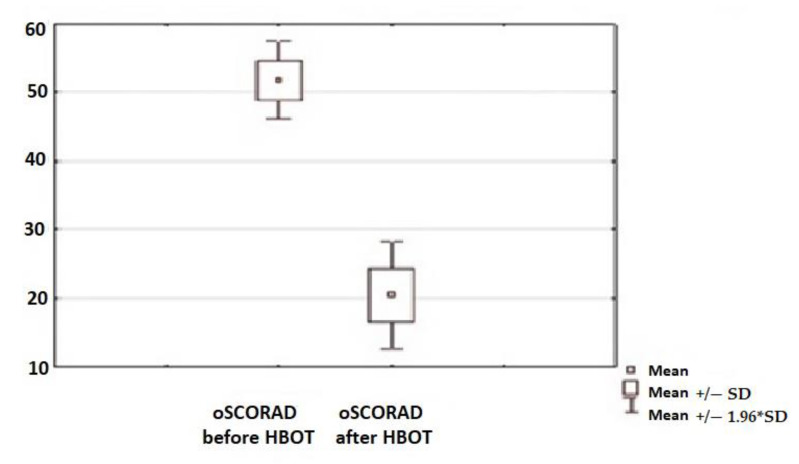
Assessment of the disease activity before and after HBOT with the use of the objective SCORAD (oSCORAD) method (*p* < 0.05).

**Figure 4 jcm-10-01157-f004:**
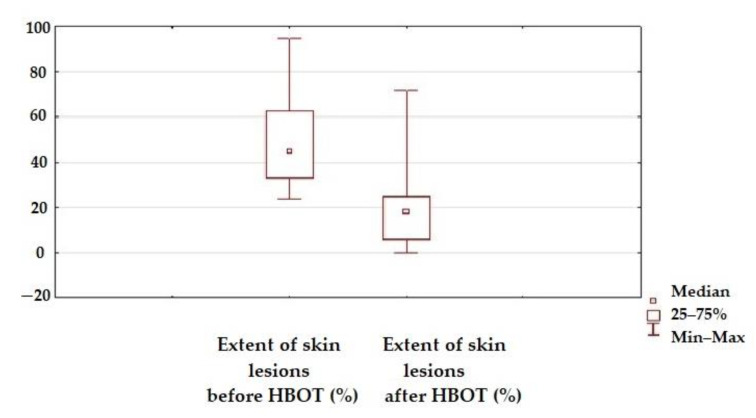
Extent of skin lesions before and after HBOT (*p* < 0.05).

**Table 1 jcm-10-01157-t001:** Inclusion and exclusion criteria for the study.

Inclusion Criteria	Exclusion Criteria
Children ≥3 years of age and <18 years of age	Children with symptoms of respiratory tract infection
Cooperative patients	Neurological contraindications (history of seizures or epilepsy)
Tolerance of confined spaces (assessment of the guardians and the team of the Clinical Department of Hyperbaric Medicine)	Pulmonary contraindications (uncontrolled asthma, pulmonary hypertension)
Severe atopic dermatitis (>50 points on SCORAD and >40 points on the oSCORAD scale)	Otolaryngological contraindications
Elevated serum total IgE above age norm	Cardiological contraindications (heart defects, hemodynamically significant PFO)
No improvement after the use of available treatment methods	Oncological contraindications (immunosuppressive treatment for neoplastic disease)
Written consent of the legal guardians/patient (if they are above 16 years old)	

**Table 2 jcm-10-01157-t002:** Characteristics of the study group.

**Age (Years)**	**(Mean ± SD)**	**10.4 ± 4.4**
**Sex (Girls/Boys)**		7:8
**Weight (percentiles)**	**(Mean ± SD)**	57 ± 20
**Height (percentiles)**	**(Mean ± SD)**	52 ± 29
**Serum total IgE antibodies (IU/mL)**	**(Median, Q1, Q3)**	1803 (Q1, 1000; Q3, 9000)
**SCORAD points**	**(Mean ± SD)**	66.2 ± 15.1
**oSCORAD points**	**(Mean ± SD)**	51.7 ± 11.2
**WBC (×10^9^/L)**	**(Mean ± SD)**	7.9 ± 2.3
**Eosinophils (×10^9^/L)**	**(Mean ± SD)**	0.6 ± 0.5
**Eosinophils (%)**	**(Mean ± SD)**	8.1 ± 5.6
**CRP (ng/dl)**	**(Mean ± SD)**	0.1 ± 0.05
**Duration of the disease (Years)**	**(Mean ± SD)**	8.8 ± 4.2
**Positive family history of allergies**		2 patients
**Coexisting atopic diseases**		1 patient: asthma
		2 patients: allergic rhinitis
		12 patients: inhaled and/or food allergy
**Antihistamines used**		11 patients
SCORAD—scoring atopic dermatitis; oSCORAD—objective scoring atopic dermatitis; WBC—white blood cells		

**Table 3 jcm-10-01157-t003:** Effect of the HBOT on the skin’s clinical condition (0–3 scale), the intensity of pruritus and sleep disorders (median, numeric scale: 0–10 points), *p* < 0.05.

SCORAD Objective Parameters	Before HBOT Median, *n* = 15	After HBOT Median, *n* = 15	*p*
Skin dryness	2	1	0.0400
Redness	2	1	0.0090
Swelling	1	0	0.0200
Oozing/crusting	2	1	0.0005
Scratch marks	2	1	0.0020
Skin lichenification	2	1	0.0002
**SCORAD subjective parameters**	**Before HBOT** **median, *n* = 15**	**After HBOT** **median, *n* = 15**	***p***
Pruritus	9	4	0.030
Sleep disorders	7	3	0.040

**Table 4 jcm-10-01157-t004:** Effect of the HBOT on immunological parameters (measured in the blood serum).

Immunological Parameter	Before HBOT			After HBOT			*p*
	Q1	Median	Q3	Q1	Median	Q3	
**Total serum IgE (IU/mL)**	1000	1803	9000	1000	1661	6000	0.002
**IL-4 (pg/mL)**	2.5	3.83	4.2	1.8	2.98	4	1
**IL-6 (pg/mL)**	1.9	1.93	3.6	1.7	1.65	2.2	0.26
**IL-10 (pg/mL)**	1.4	1.99	3.4	1.6	2.88	3.6	0.42
**Treg CD4 + CD25highCD127-FOXP3+ (%)**	0.7	0.98	1.4	0.55	0.71	0.9	0.14
**NKT (%)**	1.8	2.01	3.5	1.5	1.71	3.8	0.6

## Data Availability

The study did not report any data.

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
