# Peer review of "Effects of Hyperbaric Oxygen Therapy in Children with Severe Atopic Dermatitis"

_jcm, 2021, doi:10.3390/jcm10061157_

Round 1

Reviewer 1 Report

The Authors conducted an interesting study to assess the effects of hyperbaric oxygen therapy treatment for severe AD in children. The topic of the manuscript is relevant.

There are few issues that should be addressed:

1) The results may be influenced by the previous systemic treatment with csa or corticostroids. Did the Authors consider a washout? If not it should be mentioned as a possible limitation

2) Limitations: open, uncontrolled and unrandomized study

3) The timing of SCORAD assessment and blood sample collection (before and after HBOT) is not clear. 

Minor issues:

4) median (Q1, Q3) could be also expressed as IQR

Author Response

We are very grateful for your valuable comments on the study. Obviously, the necessary changes will be included in the final version of the article.
Point 1: The results may be influenced by the previous systemic treatment with csa or corticostroids. Did the Authors consider a washout? If not it should be mentioned as a possible limitation.
Response 1: During HBOT, only the topical treatment was used (skin care with emollients). Possible systemic glucocorticotherapy or CSA was discontinued at least several months before the initiation of HBOT. Thus, the use of the above medicaments in patients with a history of AD did not significantly affect the results of HBO treatment.
Point 2: Limitations: open, uncontrolled and unrandomized study
Response 2: The study was designed as prospective. It is necessary to conduct further, randomized studies in this area on a larger group of patients.
Point 3: The timing of SCORAD assessment and blood sample collection (before and after HBOT) is not clear.
Response 3: SCORAD and oSCORAD (assessment of the severity / activity of the
disease) were assessed before blood collection for laboratory tests (technically: on the same day or 24 hours before laboratory tests, always up to 24 hours before the initiation of HBOT therapy). We have added this information in the “Materials and methods” section.
Point 4: Median (Q1, Q3) could be also expressed as IQR.
Response 4: Of course median (Q1, Q3) could be also expressed as IQR. This is also a good alternative way to express the median. We will consider it in the presentation of our results.

Reviewer 2 Report

The authors described “The effects of hyperbaric oxygen therapy in children with severe atopic dermatitis”.

This article would provide better therapeutic options in children with severe atopic dermatitis.

However, there are several issues need to be addressed prior to being accepted.

Major comments:

  1. The results section is too short to be clear.

Based upon the methods described in the Materials and Methods, the results section should state the findings of your study concisely and objectively in a logical order.

It would also be advisable to provide a brief conclusion in each section.

  1. In contrast to the results section, the discussion is too long.

I suggest the authors to make it concise because the readers often lose sight of the main message.

The immunological parameters shown in Table 4 are not sufficient to discuss the molecular mechanisms underlying the immunomodulatory effects of HBOT.

Given that itching is the hallmark of atopic dermatitis, a focus should also be drawn to the recent findings that IgE-R+ basophil-neuronal interactions trigger for AD itch flares, and also AD chronic itch is mediated by TSLP-elicited basophils and sensory neurons activated by IL-13 or IL-31.

Author Response

We are very grateful for your valuable comments on the study. Obviously, the necessary changes will be included in the final version of the article.

Point 1: The results section is too short to be clear. Based upon the methods described in the Materials and Methods, the results section should state the findings of your study concisely and objectively in a logical order. It would also be advisable to provide a brief conclusion in each section.

Response 1: As for the discussion and the results section, we will consider this comment and we will try this part of the study to make it more readable.

Point 2: In contrast to the results section, the discussion is too long. I suggest the authors to make it concise because the readers often lose sight of the main message. The immunological parameters shown in Table 4 are not sufficient to discuss the molecular mechanisms underlying the immunomodulatory effects of HBOT.

Given that itching is the hallmark of atopic dermatitis, a focus should also be drawn to the recent findings that IgE-R+ basophil-neuronal interactions trigger for AD itch flares, and also AD chronic itch is mediated by TSLP-elicited basophils and sensory neurons activated by IL-13 or IL-31.

Response 2: We agree that immunological parameters presented in Table 4 are not sufficient to discuss the molecular machanisms underlying the immunomodulatory effects of HBOT. The study focused mainly on the analysis of the parameters involved in the inflammatory reaction in AD and the possible influence of HBOT on the percentage of TregFOXP3 lymphocytes involved in "extinguishing" the excessive immune reaction.

Subsequent studies should evaluate the interaction of IgE-R with neuronal basophils and other immunological parameters, e.g. IL-13 or IL-31, in order to better understand the pathogenesis of the disease. It was not the subject of our research.

The small number of children undergoing therapy did not allow for a broader assessment of the dependence of the examined parameters and the establishment of more binding conclusions. In our study, no comparative analysis of the assessed immunological parameters in the affected skin was conducted with the results obtained from the peripheral blood of children with AD. Our study clearly showed a favorable local effect of HBOT, which is associated with a significant improvement in the quality of life of children with AD.

Round 2

Reviewer 2 Report

The paper would be more suitable for publication after major revision changes according to the comments and suggestions.

Author Response

We are very grateful for your valuable comments on the study. Our answers are presented below.

Point 1: The results section is too short to be clear.

Based upon the methods described in the Materials and Methods, the results section should state the findings of your study concisely and objectively in a logical order. It would also be advisable to provide a brief conclusion in each section.

Response 1: We changed the result section and added brief conclusion in each section. Moreover, in this section we added the following sentences:

Lines 213-216: “The following coexisting atopic diseases were observed in children undergoing HBOT: allergic rhinitis (3 patients), inhalation and / or food allergy (12 patients). Two out of 15 patients had a positive family history of allergy. Due to severe itching, 11 children used antihistamines during hyperbaric sessions. All children completed a 30-day HBOT course.”

Lines 220-221: “Clinical assessment of the AD activity was assessed  before the initiation of HBO therapy and after the end of a 30-day cycle of the therapy.

We moved the text: “There was also a reduction in skin dryness, erythema severity, the presence of edema, exudates, scratch marks and skin lichenification. Patients also reported the reduction in the intensity of pruritus and improvement in sleep quality after HBOT” from lines 227- 229 to lines 233-236.

Lines 240-242: “In conclusion, after a 30-day cycle of HBO treatment, a beneficial clinical improvement in the skin condition was observed. There was also a reduction in the intensity of pruritus and improvement in sleep quality in all patients after HBOT.

In lines 252-257 we added a brief conclusion: “In conclusion, there was no significant influence of HBOT on the concentrations of IL-4, IL-6, IL-10 cytokines in the blood serum in examined children before and after the therapy. There was no evidence of a change in the profile of cytokines production after the treatment in a hyperbaric chamber. There were no changes in the percentages of the NKT and natural Treg cells (CD4+CD25highCD127-FOXP3+ phenotype) in the peripheral blood of patients with severe AD before and after the a 30-day cycle of HBO therapy.”

Point 2: In contrast to the results section, the discussion is too long.

I suggest the authors to make it concise because the readers often lose sight of the main message.

Response 2:

We have shortened the discussion to make it more readable.

Point 3: The immunological parameters shown in Table 4 are not sufficient to discuss the molecular mechanisms underlying the immunomodulatory effects of HBOT.

Given that itching is the hallmark of atopic dermatitis, a focus should also be drawn to the recent findings that IgE-R+ basophil-neuronal interactions trigger for AD itch flares, and also AD chronic itch is mediated by TSLP-elicited basophils and sensory neurons activated by IL-13 or IL-31.

Response 3:

We are aware that immunological parameters presented in Table 4 are not sufficient to discuss the molecular machanisms underlying the immunomodulatory effects of HBOT. The study focused mainly on the analysis of the parameters involved in the inflammatory reaction in AD and the possible influence of HBOT on the percentage of TregFOXP3 lymphocytes involved in "extinguishing" the excessive immune reaction. In discussion we have added information in the text on the recent discovery that IgE-R + basophils-neuronal interactions are a source of acute itching in AD (Lines 385-389).

All changes, pertaining to the reviewers’ suggestions are marked in the track system in the revised manuscript. The new version of the text was English edited by the native English speaker.
